# Inflammatory and Redox Responses During Medical Treatment of Open-Cervix Pyometra in Female Dogs: A Prospective Study

**DOI:** 10.3390/ani15243531

**Published:** 2025-12-08

**Authors:** Luana de Sousa Rocha, Juliana Sanches Nakaya, Roberto Rodrigues da Rosa Filho, Maria Claudia Pereda Francischini, Marcella Araujo Cebim, Thalita Farias Santos, Camila Infantosi Vannucchi

**Affiliations:** Department of Animal Reproduction, School of Veterinary Medicine and Animal Sciences, University of São Paulo, Rua Prof. Dr. Orlando Marques de Paiva 87, São Paulo 05508-270, SP, Brazil; luanarocha31@hotmail.com (L.d.S.R.); juliana_sanches@usp.br (J.S.N.); betorrf@usp.br (R.R.d.R.F.); claudiapereda@usp.br (M.C.P.F.); marcella_cebim@hotmail.com (M.A.C.); thalitafariass.vet@gmail.com (T.F.S.)

**Keywords:** acute phase proteins, oxidative stress, lipid peroxidation, cloprostenol, aglepristone

## Abstract

Pyometra is one of the most important uterine diseases in dogs, represented by infectious and inflammatory disturbance of the uterus. Thus, in therapeutic management, it is essential to ensure elimination of uterine content, as well as oxidative stress, and inflammation control. We aimed to evaluate blood redox and acute phase proteins throughout two pharmacological treatment protocols for pyometra bitches. Ten bitches with open-cervix pyometra were treated over 8 days with either aglepristone (*n* = 5) as a monotherapy or in combination with daily injections of prostaglandin (*n* = 5). Blood samples were collected daily to evaluate the concentration of liver enzymes, acute phase proteins, and the oxidative profile. In the aglepristone group, there was an increase in albumin and superoxide dismutase, while protein oxidation and reduced glutathione decreased progressively throughout treatment. The aglepristone + prostaglandin group had lower alanine aminotransferase levels on day 4 and 8 but higher lipid peroxidation, glutathione peroxidase, and C-reactive protein from the 6th to 8th day of treatment. No significant differences were detected for haptoglobin and serum amyloid A. In conclusion, oxidative markers, antioxidant enzymes, and C-reactive protein are not indicated to assess treatment efficacy when prostaglandin is employed. Conversely, albumin is a sensitive marker of therapeutic evolution for both protocols for pyometra in bitches.

## 1. Introduction

Pyometra is a uterine disease of bacterial origin that is highly prevalent in the canine species due to the particular characteristics of the estrous cycle. The estrogenic activity during estrus, followed by prolonged diestrus exposure of the endometrium to progesterone, induces structural alterations that predispose the uterine environment to bacterial colonization [1,2]. Under these conditions, the uterus becomes favorable to the proliferation of opportunistic bacteria, with *Escherichia coli* being the most frequently isolated pathogen [3].

Bacterial infection occurs via an ascending route from the vagina, facilitated by progesterone, which enhances bacterial adhesion to the endometrial epithelium and promotes bacterial biofilm formation [4,5]. The release of lipopolysaccharides (LPSs) by *E. coli* may impair neutrophil and macrophage activity, leading to dysregulation of proinflammatory cytokine production and excessive release of secondary inflammatory mediators, such as reactive oxygen species (ROS), thereby exacerbating the systemic inflammatory response [6].

As an inflammatory and infectious uterine condition, pyometra triggers systemic repercussions, including increased oxidative stress [7] and a heightened inflammatory response [8]. Indeed, once the disease is established, there is an upregulation of acute phase proteins (APPs), such as C-reactive protein (CRP) [8]. In response to inflammatory stimuli, APPs can be classified as negative (e.g., albumin) or positive (e.g., CRP, haptoglobin, and serum amyloid A-SAA) [9]. Beyond serving as inflammatory biomarkers, certain proteins also act as non-enzymatic antioxidants [10], contributing to the protection of cellular structures against oxidative damage induced by ROS and reactive nitrogen species (RNS) [11]. In addition, enzymatic antioxidants—including superoxide dismutase (SOD), catalase, reduced glutathione (GSH), and glutathione peroxidase (GPx)—play a key role in neutralizing oxidative stress [12].

The treatment of choice for pyometra is ovariohysterectomy (OHE), which provides a rapid and safe resolution, particularly in cases with severe systemic involvement [13]. However, medical alternatives may be preferred when surgery is contraindicated or when preservation of fertility is desired [14,15]. The use of uterotonic agents and antiprogestins has been shown to promote clinical improvement, reduction in infectious markers, restoration of uterine hemodynamics, and enhanced antioxidant responses through increased nitric oxide production [16,17]. Nevertheless, the effects of pharmacological protocols—particularly those employing antiprogestins (e.g., aglepristone) alone or in combination with uterotonics (e.g., PGF_2_α)—on oxidative and inflammatory profiles in bitches with pyometra remain poorly understood. Thus, we hypothesize that medical treatments for canine pyometra restore uterine inflammation and oxidative stress, in addition to greater efficacy of the combined treatment of aglepristone and prostaglandin compared to isolated aglepristone therapy.

In this context, the present study aimed to evaluate the blood redox status and acute phase protein profile throughout two pharmacological treatment protocols for pyometra in bitches, employing aglepristone either as a monotherapy or in combination with prostaglandin.

## 2. Materials and Methods

The current study was approved by the Bioethics Committee of the School of Veterinary Medicine and Animal Science—University of São Paulo, under protocol 7790240920.

As described in detail previously, using the same study population [17], a prospective randomized and observational study was conducted in 10 open-cervix pyometra bitches, aged 4–10 years, of distinct breeds and body weights (Appendix A). To assure the appropriate sample size, an analysis was conducted with the SAS version 9.3 Power and Sample Size 12 (SAS Institute Inc., Cary, NC, USA). A retrospective analysis of the hemodynamic ultrasonographic evaluation of the uterine artery (Doppler resistance index) indicated there was a power of 0.97, which is considered an acceptable statistical power (at least 0.8). Hence, a minimum of 5 dogs per group were sufficient to demonstrate significant differences in the data. As inclusion criteria, all bitches had to have sanguineous and/or purulent vaginal discharge and at least one of the following clinical signs presumptive of pyometra: lethargy, inappetence, depression, or mild to moderate dehydration (Appendix A). Clinical signs of severe systemic conditions (acute kidney and liver dysfunction, sepsis, or peritonitis) or closed cervix pyometra were considered as exclusion criteria.

The bitches presented ultrasonographic findings compatible with pyometra: intraluminal uterine content, increased endometrial vascularization, high blood flow, and low resistance of the uterine artery [18], characterized by a low pulsatility index and resistance index in the Doppler ultrasonographic examination [19]. Subsequently, blood samples were collected in order to perform kidney (blood urea nitrogen and creatinine) and liver (total protein, albumin, alanine transaminase, and alkaline phosphatase) function tests (Appendix A). The bitches were randomly assigned to two experimental groups, according to the therapeutic protocol preconized and adapted by Fieni [20]:

Aglepristone group (*n* = 5): Bitches were subjected to subcutaneous injections of 10 mg/kg of aglepristone RU46534 (Alizin-Virbac, Carros, France) on days 1, 2, and 8 after pyometra diagnosis.

Aglepristone + prostaglandin group (*n* = 5): Bitches were subjected to injections of 10 mg/kg of aglepristone (Alizin-Virbac, Carros, France) on days 1, 2, and 8, coupled with daily intramuscular injections of 1 μg/kg of cloprostenol (Ovolute—7.5 mg/100 mL—Drag Pharma, Santiago, Chile), from days 1 to 7 after pyometra diagnosis.

All bitches remained under hospitalization and intensive care, within a controlled environment. Support therapy (5–10 mL/kg/h of intravenous 0.9% saline solution, daily) and broad-spectrum antibiotic therapy [10 mg/Kg/day of enrofloxacin (Enromic^®^—5 mg/100 mL, Microsules Laboratory, Uruguay) and 30 mg/Kg/day of metronidazole (Endonidazol^®^ 5 mg/mL, Fresenius Kabi Brazil Ltda., Aquiraz, Brazil)] were provided from days 1 to 8.

From each bitch, jugular venous blood samples (5 mL) were collected daily from day 1 (onset of therapy) to day 8 (end of therapy). Blood was immediately transferred to tubes with EDTA anticoagulant and without anticoagulant to obtain plasma and serum samples, respectively, by centrifugation at 1500× *g* for 10 min. Samples were stored at −20 °C until processing.

### 2.1. Liver and Acute Phase Protein Analysis

For the liver and acute phase protein analysis, blood samples were collected on alternate days and at specific additional time-points (i.e., days 1, 4, 6, and 8).

Serum alanine aminotransferase (ALT), alkaline phosphatase, and albumin were quantified with 100 μL of serum using a commercial kit of an automated analyzer (Labtest^®^ on the Labtest Labmax 240, Lagoa Santa, MG, Brazil).

Plasma protein quantification was performed according to the protocol described by Lowry [21], in which 5 μL of the sample was diluted in 495 μL of 0.1 M NaOH, followed by reaction with 2.5 mL of Alkaline Copper Reagent for 10 min at room temperature. After 10 min, 250 μL of diluted Folin reagent was added and the samples were immediately vortexed and then left to stand for 30 min at room temperature. The absorbance of the samples was measured using a spectrophotometer (Ultrospec 3300 pro^®^, Amersham Biosciences Corp, Piscataway, NJ, USA) at a wavelength of 660 nm. The results were expressed in g of protein/dL.

Plasma CRP, haptoglobin, and SAA were measured using commercial Sandwich ELISA and Double Antibody kits (FineTest Canine CRP^®^, Canine haptoglobin^®^, and Canine SAA^®^, Wuhan, Hubei, China) on the EPOCH2 microplate reader Biotek^®^ (BioTek Instruments, Winooski, Vermont, USA), following the protocol described by the manufacturer. For haptoglobin and CRP, the test had a sensitivity of 0.188 ng/mL, with a detection range of 0.313–20 ng/mL, while for the SAA the sensitivity was 0.375 ng/mL and the range was 0.625–40 ng/mL.

### 2.2. Oxidative and Antioxidant (Redox) Analysis

In order to obtain the oxidative profile throughout pyometra treatment, antioxidant and protein oxidation was analyzed with daily samples. Only for the analysis of the marker of oxidative stress (thiobarbituric acid reactive substances—TBARSs), all serum samples (1st to 8th day of treatment) were pooled into a unique sample named “treatment”, due to technical restrictions.

The plasma activity of the antioxidant enzymes superoxide dismutase (SOD), glutathione peroxidase (GPx), and reduced glutathione (GSH), the marker of oxidative stress (TBARS), and protein oxidation were analyzed.

The determination of SOD activity was performed according to Flohe and Otting [22] by a spectrophotometer (Ultrospec 3300 Pro, Amersham Biosciences Corp, Piscataway, NJ, USA) at 550 nm, and results were expressed as U/mL. The GPx activity was measured spectrophotometrically (Ultrospec 3300 Pro, Amersham Biosciences Corp, Piscataway, NJ, USA) at a wavelength of 340 nm [23]. The results were expressed in U/mL. GSH concentration was determined spectrophotometrically [24] at 412 nm before (A) and after 15 min (B), when 5,5′-ditio-bis-(2-nitrobenzoic) acid (DTNB) was added, and difference between A and B was calculated. The protocol was validated with the use of different known proportions of GSH, and an equation was obtained: y = 0.0146 x + 0.1191. A high linear regression coefficient (r^2^ = 0.987) attested the validation of the GSH assay. Sample results were determined by comparing to the standard and validated curve.

The measurement of lipid peroxidation or TBARSs followed the protocol of Ohkawa et al. [25]. TBARSs were quantified by spectrophotometry at 532 nm (Ultrospec 3300 Pro, Amersham Biosciences Corp, Piscataway, NJ, USA). The results were compared to a standard curve previously obtained with a solution of malondialdehyde (MDA). The lipid peroxidation index was described as nanograms of TBARSs/mL of serum. Protein oxidation was determined according to the protocol described by Levine et al. [26] and Odetti et al. [27]. A total of 100 μL of the sample was added to 100 μL of 20% Trichloroacetic Acid (TCA). The mixture was then centrifuged for 10 min at 20,817× *g*, and the supernatant was discarded. Subsequently, 500 μL of 2,4-dinitrophenylhydrazine (10 mM) was added to the samples, and 500 μL of 2 M HCl was added to the respective blanks of each sample. Both were incubated for 1 h at 37 °C, with agitation every 15 min. Then, 500 μL of 20% TCA was added, the mixture was centrifuged for 5 min at 20,817× *g*, and the supernatant was discarded. A volume of 1 mL of ethyl acetate/ethanol (1:1) was added, and after 10 min, the mixture was centrifuged again at 20,817× *g*. The pellet was subsequently resuspended in 2 mL of 1 M NaOH, and the maximum absorbance was measured at 380 nm using a spectrophotometer (Ultrospec 3300 pro^®^, Amersham Biosciences Corp, Piscataway, NJ, USA). Sample results were determined by comparing to the standard and validated curve. The absorbance was measured at 380 nm by spectrophotometry.

### 2.3. Statistical Analysis

Data were evaluated using SAS System for Windows version 9.3 (SAS Institute Inc., Cary, NC, USA). Effects of experimental group, time of evaluation, and interaction between these factors were estimated by repeated measures ANOVA (SAS MIXED procedure). Differences between groups were analyzed using parametric tests considering residual normality (Gaussian distribution), variance homogeneity, and data transformations suggested by the Guided Data Analysis Procedure of the SAS System for Windows version 9.3. Results are reported as untransformed means ± SE.

Variables were also submitted to Pearson correlation analysis, considering only the group effect. Statistical differences were considered significant if *p* ≤ 0.05.

## 3. Results

As stated by Rosa Filho et al. [17] with the same study population, both treatment groups had a significant and progressive decrease in uterine area and leukocyte count during the course of therapy, indicating the effectiveness of the treatment protocol employed here for pyometra.

### 3.1. Liver and Acute Phase Protein Analysis

Throughout the experimental period, females of the aglepristone + prostaglandin group had total plasma protein within the reference range for dogs (5.3–7.6 g/dL) but a higher concentration compared with the aglepristone group on day 3 (*p* = 0.02) and day 4 (*p* = 0.004) of treatment (Figure 1a). Conversely, bitches treated with the aglepristone protocol exhibited hypoproteinemia on the 3rd and 7th days of treatment, although no significant differences were observed among evaluation time-points within either experimental group (Figure 1a).

Except for the first day of treatment with the aglepristone + prostaglandin protocol, plasma albumin concentration was within the reference range (2.3–3.8 g/dL) in both groups; however, a significant increase was observed only in the aglepristone group throughout treatment, whereas results remained unchanged in females treated with the combined protocol (Figure 1b). No differences were detected between experimental groups throughout the treatment period for plasma albumin concentration (Figure 1b).

Bitches treated with aglepristone alone exhibited lower circulating alanine aminotransferase (ALT) levels compared with the aglepristone + prostaglandin group on day 4 (*p* = 0.004) and day 8 (*p* = 0.006), although ALT concentration remained within the reference range (10–88 U/L) for both groups throughout treatment (Figure 2). No statistical differences were observed in alkaline phosphatase concentrations either between groups or across evaluation time-points (Table 1). On the other hand, aglepristone-treated bitches had alkaline phosphatase concentrations above the normal range (20–151 U/L) throughout the entire period, whereas the aglepristone + prostaglandin group remained within the normal values (Table 1).

No significant differences were detected in the concentrations of haptoglobin and serum amyloid A, either between treatment groups or among experimental time-points (Table 1). However, for the aglepristone + prostaglandin group, there was a significant increase in C-reactive protein from the 6th to the 8th day of treatment (Table 1). In addition, in both groups, CRP blood concentration was above the reference range (1–12 ng/mL) both on day 4 and 8 of the treatment protocol (Table 1). Serum concentration of amyloid A remained within the normal preconized value for healthy dogs (inferior to 5 ng/mL) in both groups during the entire treatment period (Table 1).

### 3.2. Oxidative and Antioxidant (Redox) Analysis

Females treated with the combined aglepristone and prostaglandin protocol showed significantly higher lipid peroxidation (*p* = 0.001) compared with those treated with aglepristone alone (Figure 3a). Aglepristone therapy resulted in a significantly lower level of protein oxidation (*p* = 0.01) at mid-treatment (day 4) compared with the aglepristone + prostaglandin group (Figure 3b). Furthermore, in bitches treated with aglepristone alone, protein oxidation decreased significantly on days 3 and 8 relative to the onset of treatment (day 1) (Figure 3b).

During aglepristone therapy, a significant increase in serum superoxide dismutase (SOD) activity was observed between days 4 and 7 after the onset of treatment (Figure 4). Conversely, SOD concentration in the aglepristone + prostaglandin group was higher on day 3 (*p* = 0.02) and day 4 (*p* = 0.04) compared with bitches treated with aglepristone alone (Figure 4).

No significant variation was detected in the circulating glutathione peroxidase (GPx) profile throughout pyometra treatment in either group. Nevertheless, on day 4, GPx concentration was significantly higher (*p* = 0.03) in females receiving the combined aglepristone + prostaglandin therapy (Figure 5a). In the aglepristone group, serum reduced glutathione (GSH) concentration showed a progressive decline over the course of treatment, with a significant reduction on day 7 compared to days 2–4 of the protocol (Figure 5b). No significant differences were detected across treatment days in the aglepristone + prostaglandin group or between the two experimental groups for GSH concentration (Figure 5b).

### 3.3. Correlation Analysis

For the aglepristone group, total protein concentration correlated negatively (r = −0.62; *p* = 0.02) with reduced glutathione (GSH). In addition, there was a negative correlation (r = −1.0; *p* < 0.0001) between lipid peroxidation (TBARS) and C-reactive protein.

For the aglepristone + prostaglandin group, positive correlations were observed between albumin concentration and superoxide dismutase (r = 0.58; *p* = 0.02) and with glutathione peroxidase (r = 0.72; *p* = 0.003).

## 4. Discussion

As an inflammatory condition of systemic nature, pyometra is capable of eliciting oxidative systemic responses; therefore, treatment efficacy also involves the restoration of redox balance and inflammatory homeostasis. Accordingly, the present study employed key markers of inflammation (acute phase proteins and hepatic enzymes) and oxidative metabolism to evaluate the effectiveness of two therapeutic protocols for pyometra in bitches.

It is important to mention that all females, regardless of the pyometra treatment protocol, received antibiotic support and fluid therapy, which was imperative for combating the infectious process, evidenced by the progressive decrease in leukocyte counts throughout therapy. We believe that the appropriate choice of antibiotic is of paramount importance for the ongoing therapeutic response and contributes significantly to the success of the treatment. In the aglepristone + prostaglandin group, reduced concentration of circulating albumin (hypoalbuminemia) was observed at onset of treatment, which indicates the concomitant presence of an inflammatory and infectious process, triggering a negative influence of cytokines on hepatic synthesis [28]. However, mean albumin concentration progressively increased throughout therapy, reaching values within the physiological range for the species. Our findings differ from those reported by Hadzimusic and Alagic [29], who observed no alterations in albumin levels following surgical treatment of bitches with pyometra. In contrast, Motawakkel et al. [30] demonstrated an increase in albumin concentration in bitches after medical treatment for pyometra. Thus, it may be inferred that therapeutic modalities for pyometra exert distinct effects on the blood albumin profile, as surgical trauma itself constitutes an inflammatory stimulus. Therefore, we suggest that the efficacy of medical therapy for pyometra may be monitored through serial evaluation of serum albumin, an assessment not typically indicated in surgical treatment protocols. To reinforce this assertion, we observed a positive correlation between blood concentrations of albumin (a negative inflammatory protein) and the antioxidant enzymes SOD and GPx in bitches treated with the combined protocol (aglepristone + prostaglandin), indicating that as the therapy took effect, there was a reduction in the mobilization of antioxidant enzymes, increasing their systemic availability. In other words, only in the aglepristone + prostaglandin group, the restoration of hepatic function and improvement of the inflammatory condition was accompanied by reduced antioxidant consumption to counter oxidative damage.

On the other hand, treatment with aglepristone alone dynamically influenced the superoxide dismutase profile during therapy. The lower antioxidant activity observed midway through the treatment course indicates possible enzymatic consumption in the efficient neutralization of reactive oxygen species (ROS) generated by the inflammatory process, particularly those related to superoxide radical (O_2_^−^) generation [31]. The effectiveness of SOD in counteracting oxidative stress induced by pyometra was further evidenced by a subsequent compensatory increase in enzyme concentration approaching the end of treatment (day 7), suggesting the restoration of circulating levels as oxidative damage subsides. Similar findings were reported by Yazlık et al. [7], who evaluated SOD profiles in bitches with pyometra-induced sepsis. In contrast, when prostaglandin was combined with aglepristone, higher SOD activity was observed, possibly reflecting the direct uterotonic action of cloprostenol, even at reduced doses. Therefore, in addition to the oxidative stimulus associated with the uterine disorder, prostaglandin-induced myometrial activity represents an additional factor requiring enzymatic neutralization; consequently, no oscillatory SOD consumption pattern was detected during combined therapy. Similarly to the superoxide dismutase profile, serum GPx concentrations were higher in the aglepristone + prostaglandin group on day 4 of treatment, supporting the hypothesis of an additional antioxidant response associated with the myometrial contractility stimulated by cloprostenol—specifically catalyzing the reduction of hydrogen peroxide (H_2_O_2_) or organic hydroperoxide (ROOH) to water (H_2_O) and alcohol (ROH), respectively [31]. Such antioxidant responses have been reported following surgical treatment of pyometra bitches, indicating a compensatory mechanism and restoration of post-surgical tissue homeostasis [32].

However, for GPx to exert its antioxidant function efficiently, GSH (a non-protein thiol) acts as an essential substrate [33]. In our study, a progressive decrease in serum GSH concentration was observed throughout aglepristone treatment, resulting from its continuous consumption in ROS neutralization, particularly hydrogen peroxide [34]. In fact, in the aglepristone group, there was a negative correlation between GSH serum concentration and total protein, indicating that while inflammatory proteins are less synthetized, circulating antioxidant enzymes become more available. Similarly, Szczubial et al. [35] reported lower uterine GSH concentrations in bitches with pyometra compared with healthy females.

Regarding the acute phase proteins tested in the present study, no changes were detected in the circulating profile of haptoglobin and serum amyloid-A throughout treatment in either group, which may be attributed to the selection criteria of the subjects, as only bitches with open-cervix pyometra and without secondary systemic involvement were included in this study. Indeed, hepatic synthesis of acute phase proteins depends on both the intensity of the inflammatory insult and its resolution [36,37,38]. Thus, reversal of the inflammatory process by medical treatment for pyometra does not occur abruptly enough to be detected by fluctuations in haptoglobin and SAA concentrations. Further studies involving cases of varying severity, pathogenicity, and therapeutic protocols are therefore required to elucidate the pattern of acute phase protein synthesis in canine pyometra.

On the other hand, although no difference was observed between experimental groups, C-reactive protein concentration was higher than the upper threshold value on the 4th day of treatment with the combined protocol (aglepristone + prostaglandin), confirming the ongoing inflammatory/infectious process in canine pyometra [36]. But a significant increase in CRP concentration occurred at the end of treatment (8th day) compared to the former result. We therefore hypothesize that prostaglandin administration may stimulate hepatic synthesis and release of C-reactive protein. Using this same reasoning, we observed greater activity of the liver enzyme alanine transaminase (ALT) on the 4th and 8th days of treatment in the aglepristone + prostaglandin group, reinforcing our suggestion of a possible action of prostaglandin in stimulating liver activity as a whole. In addition, Unnikrishnan et al. [39] described a 15-day delay for CRP concentrations to normalize after starting drug treatment for pyometra bitches with mifepristone. However, it is important to emphasize that a negative correlation between CRP and lipid peroxidation (TBARS) was observed in the aglepristone group, indicating the possible role of C-reactive protein as a scavenger of the oxidative stress in canine pyometra. In any case, our data raise awareness of using CRP levels as a marker of favorable inflammatory progression whenever a therapeutic protocol involving prostaglandin administration is employed.

As part of the global mechanism of oxidative stress, protein carbonyl content represents the degree of oxidation of amino acid residues in protein structures [40], whereas lipid peroxidation results from the degradation of the lipid fraction of cellular architecture by ROS, leading to plasma membrane damage [41]. In our study, a significant reduction in protein carbonyl concentration was observed on days 4 and 8 following initiation of treatment in the aglepristone group, indicating the therapeutic efficacy of this protocol in reducing oxidative damage to protein structures. Because protein carbonylation is regarded as an important marker of inflammatory disease—owing to its early formation and detectable stability in protein side chains [42,43]—we can infer that the use of aglepristone effectively reduces protein oxidation by reactive oxygen species induced by pyometra in bitches. Conversely, greater protein oxidation was detected on day 4 in the group treated with combined aglepristone and prostaglandin compared with aglepristone alone. Furthermore, bitches treated with the combined protocol exhibited higher lipid peroxidation, as indicated by TBARS concentrations. Although this oxidative profile may represent an exacerbation of redox imbalance after initiation of the combined therapy, it is noteworthy that prostaglandin F2α per se can stimulate reactive oxygen species production [44] through a mechanism independent of the NOS-NO system in the induction of lipid peroxidation [45].

## 5. Conclusions

Both therapeutic protocols resulted in clinical improvement in all animals; however, the combined use of prostaglandin modified the profile of oxidative markers, antioxidant enzymes, and C-reactive protein, thereby preventing the assessment of treatment efficacy in reversing the inflammatory and redox metabolism caused by pyometra. Conversely, the circulating profile of albumin, a negative acute phase protein, proved to be a sensitive and practical marker of therapeutic evolution in both the aglepristone protocol and when it was combined with prostaglandin for pyometra in bitches. Nevertheless, further research should be conducted with a larger number of subjects, including distinct severity manifestations and employment of methodological tools with greater sensitivity for the accurate detection of biological markers.

## Figures and Tables

**Figure 1 animals-15-03531-f001:**
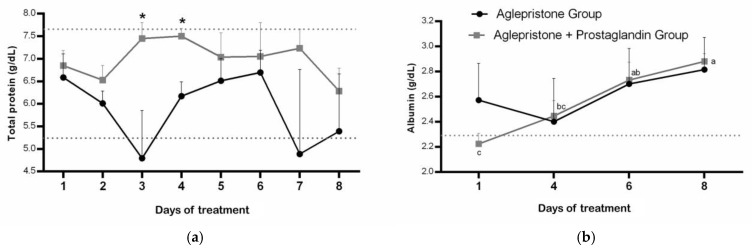
Mean ± SE of (**a**) plasma protein concentration (g/dL) and (**b**) albumin (g/dL) in the aglepristone and aglepristone + prostaglandin groups throughout the experimental time-points. ^a–c^ different letters represent significant differences (*p* < 0.05) between time-points within the same experimental group. * indicates significant differences (*p* = 0.02 and *p* = 0.004, respectively) between groups within the same time-point. Dotted lines represent the reference range for dogs.

**Figure 2 animals-15-03531-f002:**
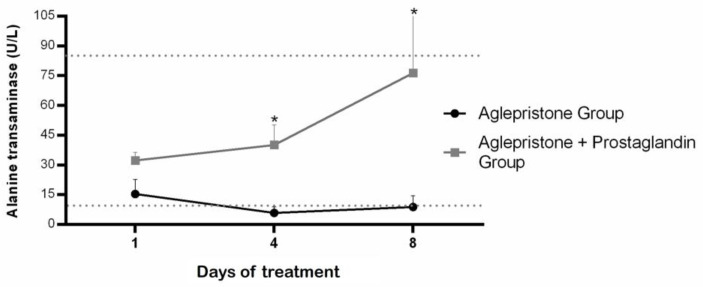
Mean ± SE of alanine transaminase (U/L) in the aglepristone and aglepristone + prostaglandin groups throughout the experimental time-points. * indicates significant differences (*p* = 0.004 and *p* = 0.006, respectively) between groups within the same time-point. Dotted lines represent the reference range for dogs.

**Figure 3 animals-15-03531-f003:**
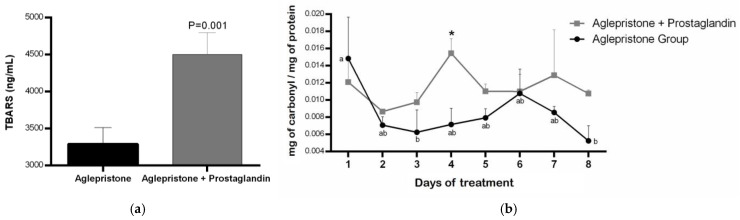
Mean ± SE of (**a**) lipid peroxidation (TBARS; ng/mL) in the aglepristone and aglepristone + prostaglandin groups and (**b**) protein oxidation (mg of carbonyl/mg of protein) throughout the experimental time-points. ^a,b^ different letters represent significant differences (*p* < 0.05) between time-points within the same experimental group. * indicates significant differences (*p* = 0.01) between groups within the same time-point.

**Figure 4 animals-15-03531-f004:**
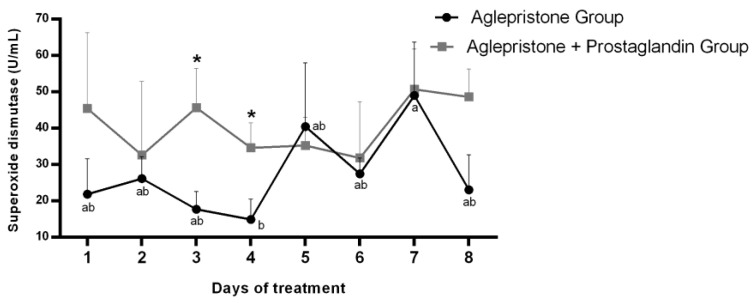
Mean ± SE of superoxide dismutase activity (U/mL) in the aglepristone and aglepristone + prostaglandin groups throughout the experimental time-points. ^a,b^ different letters represent significant differences (*p* < 0.05) between time-points within the same experimental group. * indicates significant differences (*p* = 0.02 and *p* = 0.04, respectively) between groups within the same time-point.

**Figure 5 animals-15-03531-f005:**
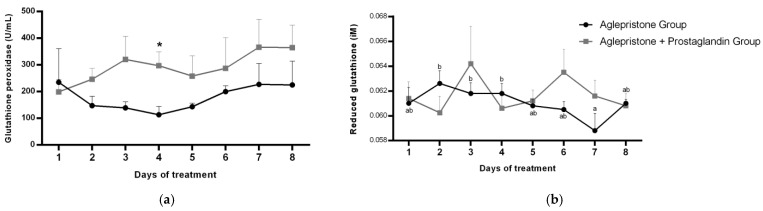
Mean ± SE of (**a**) glutathione peroxidase activity (U/mL) and (**b**) reduced glutathione (µM) in the aglepristone and aglepristone + prostaglandin groups throughout the experimental time-points. ^a,b^ different letters represent significant differences (*p* < 0.05) between time-points within the same experimental group. * indicates significant difference (*p* = 0.03) between groups within the same time-point.

**Table 1 animals-15-03531-t001:** Mean and SE of blood concentration of alkaline phosphatase (U/L), C-reactive protein (ng/mL), haptoglobin (ng/mL), and serum amyloid-A (ng/mL) throughout pyometra treatment in the aglepristone and aglepristone + prostaglandin groups.

Day of Treatment	Aglepristone Group				Aglepristone + Prostaglandin Group			
	Alkaline Phosphatase	C-Reactive Protein	Amyloid-A	Haptoglobin	Alkaline Phosphatase	C-Reactive Protein	Amyloid-A	Haptoglobin
Day 1	**248.0 ± 89.4**	7.3 ± 2.8	3.8 ± 1.4	45.5 ± 12.7	113.7 ± 40.0	9.6 ± 2.1 ^ab^	2.4 ± 1.2	52.2 ± 12.6
Day 4	**215.8 ± 94.6**	**18.8 ± 10.1**	3.1 ± 0.9	49.5 ± 3.4	115.8 ± 39.4	**18.3 ± 6.2 ^ab^**	2.5 ± 1.3	67.8 ± 16.7
Day 6	**--**	11.1 ± 6.4	2.1 ± 0.5	114.1 ± 61.7	--	9.0 ± 2.7 ^a^	2.62 ± 1.3	63.2 ± 34.2
Day 8	**169.0 ± 81.8**	**55.7 ± 50.5**	2.6 ± 0.6	155.6 ± 108.5	145.0 ± 43.7	**79.9 ± 41.4 ^b^**	2.3 ± 1.0	416.9 ± 369.2

Bold values represent results outside the normal range for dogs. a–b different letters represent significant differences (*p* < 0.05) between time-points within the same experimental group

## Data Availability

Dataset available on request from the authors.

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
