# Peer review of "Inflammatory and Redox Responses During Medical Treatment of Open-Cervix Pyometra in Female Dogs: A Prospective Study"

_animals, 2025, doi:10.3390/ani15243531_

Round 1
Reviewer 1 Report
Comments and Suggestions for Authors
Nicely written article, here are some small comments and suggestions.
Line 11 – replace endometrium by uterus
Line 45 – since the outcome of the 10 bitches was positive (successful result in the 10 cases), is it not better to talk about therapeutic effectiveness than evolution?
Line 94 – I think the population should be well defined in the article, the reader should not need to read another paper to find out the population demographic presented in the article.
Line 98 – Since for both cases (line 94 and 98) the reference is the same, I think it will be easier to understand if you explain directly how sample size was calculated.
Line 111 – therapeutic protocol preconised and adapted by Fieni (20).
Line 114 – delete, in accordance…
Line 118 – delete (protocol adapted…
Line 155 – The definition needs to be presented before the acronym for TBARS (line 158)
Line 161 – replace “the assay was performed in” for by
Line 171 –172 - once the definition of the acronym is done, there is no reason to redo it
Paragraph 365-381 – check the abbreviations CPR and CRP
Author Response
November 27th, 2025
Dear Reviewer,
We want to thank the opportunity of submitting our manuscript to this prestigious journal and greatly appreciate all the efforts spent on reviewing our work. We are certain that your contributions greatly improved the quality of this manuscript. We hope our modifications make it suitable for acceptance.
We have accepted all your suggestions. Some interesting points were raised, which certainly make the text clearer and easier to understand. Subsequently, we provide a point-by-point description of the modifications made:
The suggestions have been addressed and written in RED in the revised manuscript.
Line 11 – replace endometrium by uterus
The term was replaced accordingly in Line 12.
Line 45 – since the outcome of the 10 bitches was positive (successful result in the 10 cases), is it not better to talk about therapeutic effectiveness than evolution?
The sentence was rephrased accordingly in Line 45.
Line 94 – I think the population should be well defined in the article, the reader should not need to read another paper to find out the population demographic presented in the article.
We apologize for the lack of clarity. The exact description of the study population was provided in Table 1 of the supplementary material (Table 1 S1, Table 2 S1 and Table 3 S1).
Line 98 – Since for both cases (line 94 and 98) the reference is the same, I think it will be easier to understand if you explain directly how sample size was calculated.
We agree with the reviewer and, thus, the information was properly provided in Lines 99-105.
Line 111 – therapeutic protocol preconised and adapted by Fieni (20).
The suggestion was properly accepted in Line 117
Line 114 – delete, in accordance…
The suggestion was properly accepted in Line 120.
Line 118 – delete (protocol adapted…
The suggestion was properly accepted in Line 124.
Line 155 – The definition needs to be presented before the acronym for TBARS (line 158)
The proper definition was included in Line 160.
Line 161 – replace “the assay was performed in” for by
The suggestion was properly accepted in Lines 166-167.
Line 171 –172 - once the definition of the acronym is done, there is no reason to redo it
The suggestion was properly accepted in Line 176.
Paragraph 365-381 – check the abbreviations CPR and CRP
We apologize for the typographical error, and the full text has been properly reviewed.
We are submitting the revised manuscript to be revaluated for publication.
Thank you for your consideration.
Sincerely,
Camila Infantosi Vannucchi, PhD
Professor, Department of Animal Reproduction - University of São Paulo

Reviewer 2 Report
Comments and Suggestions for Authors
Dear Authors,
I would like to congratulate you on submitting a very interesting and relevant manuscript addressing the inflammatory and redox responses during medical treatment of open-cervix pyometra in female dogs. The topic is clinically meaningful, the study is well motivated, and your work contributes valuable information to the understanding of oxidative and inflammatory dynamics during pharmacological therapy. Your effort in collecting daily biochemical data and in monitoring multiple markers is commendable.
To further strengthen the clarity, scientific rigor, and overall impact of the manuscript, I would kindly suggest the following revisions:
- The title is understandable but grammatically incorrect: “Inflammatory and redox response to conservative treatment of open-cervix pyometra bitches” is missing articles, and it should avoid using “bitches” in the title for general readership. Suggested improved title: “Inflammatory and Redox Responses During Medical Treatment of Open-Cervix Pyometra in Female Dogs: A Prospective Study”
- Several grammatical errors (e.g., "evaluated" → "to evaluate").
- No explanation of why oxidative stress markers matter clinically, and it does not state the main clinical implication or how results help veterinarians (why does albumin matter? Why are oxidative markers not useful with prostaglandin? ).
- Line 133: "blood samples were collected every other day (i.e., days 1, 4, 6, and 8...)" is incorrect. Better: “Blood samples were collected on alternate days and at specific additional time points (i.e., days 1, 4, 6, and 8).”
- “CPR” spelled inconsistently with “CRP”.
- Mention the study design: prospective observational.
- Add a sentence acknowledging sample size.
- Clarify the difference between statistical and clinical significance.
- Provide a clear hypothesis, not only an aim of the present paper.
- Should state that the power analysis is based on a different study (Rosa Filho et al.), which used hemodynamic endpoints, not oxidative markers.
- No blinding was reported; outcome assessment may be biased.
- The LSD test for time points is incorrect; mixed-effects models should be used. It is inappropriate for small-sample repeated measures.
- Please explain why TBARS was evaluated by pooling all samples across days into one sample per animal. It destroys temporal resolution and violates statistical independence.
- Add justification for the doses and timing of cloprostenol.
- Multiple P-values are missing exact numbers (e.g., “P < 0.05” is too generic).
- Figures do not include confidence intervals or individual data points, which are important for n=5. Y-axis units are missing in some descriptions.
- All animals received antibiotics and fluid therapy, which strongly influence inflammatory and oxidative markers but are not discussed.
- Statements like “prostaglandin induces liver activity as a whole” lack citation and biological basis.
- Avoid speculative statements without evidence.
- For conclusions: “Aglepristone, alone or with prostaglandin, resulted in clinical improvement in all animals; however, due to the limited sample size and methodological constraints, the findings regarding oxidative and inflammatory markers should be interpreted with caution.”
- The abbreviations list is not complete!
-
For Supplementary File (Table S1):
-
Very inconsistent formatting (decimal points, commas): Lhasa Apso 3,6 kg” → should consistently use “3.6 kg;
-
Missing units for hematology (e.g., low haemoglobin but no values);
- Should include reference ranges for clarity;
- Data indicates wide heterogeneity between animals, which should be acknowledged.
-
1. UK-English inconsistencies (e.g., "behavior," "defense" → "behaviour," "defence").
2. Suggested rephrasing: “Therapeutic approach must guarantee...” -> “In therapeutic management, it is essential to ensure…”
3. “In the Aglepristone Group, there was increase in albumin concentration…”
-> “In the Aglepristone Group, there was an increase…”
4. “Aglepristone therapy induced a significant lower level of protein oxidation”
-> “Aglepristone therapy resulted in a significantly lower level of protein oxidation…”
5. “prostaglandin activity stimulates hepatic synthesis” -> “prostaglandin administration may stimulate hepatic synthesis…”
6.
Author Response
November 27th, 2025
Dear Reviewer,
We want to thank the opportunity of submitting our manuscript to this prestigious journal and greatly appreciate all the efforts spent on reviewing our work. We are certain that your contributions greatly improved the quality of this manuscript. We hope our modifications make it suitable for acceptance.
We have accepted all your suggestions. Some interesting points were raised, which certainly make the text clearer and easier to understand. Subsequently, we provide a point-by-point description of the modifications made:
The suggestions have been addressed and written in BLUE in the revised manuscript.
The title is understandable but grammatically incorrect: “Inflammatory and redox response to conservative treatment of open-cervix pyometra bitches” is missing articles, and it should avoid using “bitches” in the title for general readership. Suggested improved title: “Inflammatory and Redox Responses During Medical Treatment of Open-Cervix Pyometra in Female Dogs: A Prospective Study”
We are grateful for the suggested title for the scientific work and have incorporated the contribution.
Several grammatical errors (e.g., "evaluated" → "to evaluate").
We apologize for the grammatical and orthographical errors, and the full text has been properly corrected.
No explanation of why oxidative stress markers matter clinically, and it does not state the main clinical implication or how results help veterinarians (why does albumin matter? Why are oxidative markers not useful with prostaglandin? ).
We are sorry for the lack of clarity with the overall goal of this research. We believe that the statement in Lines 302-305 can justify the use of oxidative stress and inflammatory markers.
Line 133: "blood samples were collected every other day (i.e., days 1, 4, 6, and 8...)" is incorrect. Better: “Blood samples were collected on alternate days and at specific additional time points (i.e., days 1, 4, 6, and 8).”
The phrase was corrected accordingly in Lines 138-139.
CPR” spelled inconsistently with “CRP”.
We apologize for the typographical error, and the full text has been properly reviewed.
Mention the study design: prospective observational
The information on the study design was provided in Lines 97-98.
Add a sentence acknowledging sample size
We agree with the reviewer and, thus, the information was properly provided in Lines 101-102.
Clarify the difference between statistical and clinical significance.
Although clinical assessment was part of the experimental procedure, we chose to rely on objective and measurable experimental variables, such as ultrasound-based uterine biometry and leukocyte counts. A clarification was provided in Lines 206-207.
Provide a clear hypothesis, not only an aim of the present paper
We apologize for the mistake; thus, the working hypothesis was included in Lines 86-89.
Should state that the power analysis is based on a different study (Rosa Filho et al.), which used hemodynamic endpoints, not oxidative markers.
We totally agree with the reviewer, therefore the statement on how de power analysis was performed was provided in Lines 101-102.
No blinding was reported; outcome assessment may be biased
Although we fully agree with the reviewer, we believe that a blinded treatment approach would be unfeasible for this specific experiment, due to the notable differences in patient management protocols for each treatment. Therefore, although clinical assessment was part of the experimental procedure, we chose to rely on objective and measurable experimental variables, such as ultrasound-based uterine biometry and leukocyte counts.
The LSD test for time points is incorrect; mixed-effects models should be used. It is inappropriate for small-sample repeated measures.
We truly apologize for the inaccuracy regarding the statistical procedure. We wish to clarify that the analysis of repeated measures over time was indeed conducted using the Mixed model in SAS. This corrected and accurate information has now been incorporated into Lines 195-197.
Please explain why TBARS was evaluated by pooling all samples across days into one sample per animal. It destroys temporal resolution and violates statistical independence.
Unfortunately, there was insufficient blood volume to perform daily determinations, as the assay required a serum volume greater than what we had available. Thus, we opted to pool all samples into a single analysis so that the evaluation could be performed. A clarification was provided in Lines 160-161.
Add justification for the doses and timing of cloprostenol.
The therapeutic protocol was the one described by Fieni (2006) as stated in Line 117
Multiple P-values are missing exact numbers (e.g., “P < 0.05” is too generic)
All probability values (P) were included in the figure captions for better clarity.
Figures do not include confidence intervals or individual data points, which are important for n=5. Y-axis units are missing in some descriptions.
We sincerely thank the reviewer for their suggestion, which is highly appreciated. Nevertheless, when simulating the suggested graphical modification, we found that the illustration became overly complex, obscuring the interpretation of the variable profiles within the experimental groups. We have included an example of the figure with individual data points below for the reviewer's consideration. Should the reviewer still deem this modification pertinent, we are willing to adjust all remaining figures.
All animals received antibiotics and fluid therapy, which strongly influence inflammatory and oxidative markers but are not discussed
We agree with the reviewer and thus a discussion on that topic was provided in Lines 306-311.
Statements like “prostaglandin induces liver activity as a whole” lack citation and biological basis.
We apologize for our writing presenting imperative statements, and thus, we have modified Lines 383-384.
Avoid speculative statements without evidence
We have reviewed the discussion in order to denote only a scientific inference and potential possibilities for biological explanations, as for example in Lines 388, 389-390 and 400.
For conclusions: “Aglepristone, alone or with prostaglandin, resulted in clinical improvement in all animals; however, due to the limited sample size and methodological constraints, the findings regarding oxidative and inflammatory markers should be interpreted with caution.”
We are grateful for the reviewer's suggestion, which was incorporated in a modified form within the Conclusion (Lines 412-422).
The abbreviations list is not complete!
The abbreviation list was revised and adjusted accordingly.
For Supplementary File (Table S1):
- Very inconsistent formatting (decimal points, commas): Lhasa Apso 3,6 kg” → should consistently use “3.6 kg;
- Missing units for hematology (e.g., low haemoglobin but no values);
- Should include reference ranges for clarity;
- Data indicates wide heterogeneity between animals, which should be acknowledged.
We apologize for missing information and formatting errors. The supplementary file was redone to include all clinical and laboratory information for the bitches in each experimental group. It was necessary to divide the table into 3 so that the maximum amount of information could be included.
Comments on the Quality of English Language
- UK-English inconsistencies (e.g., "behavior," "defense" → "behaviour," "defence").
- Suggested rephrasing: “Therapeutic approach must guarantee...” -> “In therapeutic management, it is essential to ensure…”
- “In the Aglepristone Group, there was increase in albumin concentration…”
-> “In the Aglepristone Group, there was an increase…”
- “Aglepristone therapy induced a significant lower level of protein oxidation”
-> “Aglepristone therapy resulted in a significantly lower level of protein oxidation…”
- “prostaglandin activity stimulates hepatic synthesis” -> “prostaglandin administration may stimulate hepatic synthesis…”
We apologize for the grammatical and orthographical errors, and the full text has been properly corrected. All suggestions were incorporated to the text.
We are submitting the revised manuscript to be revaluated for publication.
Thank you for your consideration.
Sincerely,
Camila Infantosi Vannucchi, PhD
Professor, Department of Animal Reproduction - University of São Paulo

Round 2
Reviewer 2 Report
Comments and Suggestions for Authors
This paper's quality has improved significantly, and I would like to take this opportunity to congratulate those involved in this endeavor. My response is to accept the article in its latest form.